# Predicting Current Potential Distribution and the Range Dynamics of *Pomacea canaliculata* in China under Global Climate Change

**DOI:** 10.3390/biology11010110

**Published:** 2022-01-10

**Authors:** Yingxuan Yin, Qing He, Xiaowen Pan, Qiyong Liu, Yinjuan Wu, Xuerong Li

**Affiliations:** 1Department of Parasitology, Zhongshan School of Medicine, Sun Yat-sen University, Guangzhou 510080, China; yinyx23@mail2.sysu.edu.cn (Y.Y.); heqing25@mail2.sysu.edu.cn (Q.H.); panxw7@mail2.sysu.edu.cn (X.P.); 2Key Laboratory for Tropical Diseases Control, Ministry of Education, Sun Yat-sen University, Guangzhou 510080, China; 3Provincial Engineering Technology Research Center for Biological Vector Control, Guangzhou 510080, China; 4China Atomic Energy Authority Center of Excellence on Nuclear Technology Applications for Insect Control, Beijing 100048, China; 5State Key Laboratory of Infectious Diseases Prevention and Control, National Institute for Communicable Disease Control and Prevention, Chinese Center for Disease Control and Prevention, Beijing 102206, China; liuqiyong@icdc.cn

**Keywords:** MaxEnt, *Pomacea canaliculata*, biological invasion, dispersal risk

## Abstract

**Simple Summary:**

*Pomacea canaliculata* is one of the 100 worst invasive alien species in the world, causing significant effects and harm to native species, ecological environment, human health, and social economy. In this study, we used species distribution modeling (SDM) methods to predict the potential distribution of *P. canaliculata* in China and found that with climate change, there would be a trend of expanding and moving northward in the future.

**Abstract:**

*Pomacea canaliculata* is one of the 100 worst invasive alien species in the world, which has significant effects and harm to native species, ecological environment, human health, and social economy. Climate change is one of the major causes of species range shifts. With recent climate change, the distribution of *P. canaliculata* has shifted northward. Understanding the potential distribution under current and future climate conditions will aid in the management of the risk of its invasion and spread. Here, we used species distribution modeling (SDM) methods to predict the potential distribution of *P. canaliculata* in China, and the jackknife test was used to assess the importance of environmental variables for modeling. Our study found that precipitation of the warmest quarter and maximum temperature in the coldest months played important roles in the distribution of *P. canaliculata*. With global warming, there will be a trend of expansion and northward movement in the future. This study could provide recommendations for the management and prevention of snail invasion and expansion.

## 1. Introduction

Heightened connectivity between countries brought about by globalization’s facilitation has contributed to tremendous economic and social development through global trade, international travel/tourism, etc. and also resulted in the introduction of numerous invasive alien species, posing significant threats to native species, the ecological environment, human health, and the social economy [1].

*Pomacea canaliculata* (Gastropoda: Ampullariidae), commonly called the apple snail, a freshwater snail native to tropical and temperate South America, was listed by the International Union of Conservation of Nature (IUCN) in the “100 of the world’s worst invasive alien species” [2], as well as among the first batch of invasive alien species in China. The invasion of *P. canaliculata* severely harmed the biodiversity in China, altered the spatial distribution of native species, caused direct harm to the production of agriculture, forestry, animal husbandry, and fishery, and resulted in massive economic losses [3]. The species can also lead to disease spread and pose serious threats as pathogen vectors to China’s public health security [4].

Some species of *Pomacea* have characteristics that have been linked to invasiveness. After invading the United States, Japan, the Philippines, and other countries, it caused serious damage to the local nature and agriculture, but the control effect of various countries was not ideal, and the population and spread area continued to increase [5,6]. *P. canaliculata*, which is widely distributed in China’s south of the Yangtze River, is the most common *Pomacea* spp. Found in the area [7,8]. Due to its omnivorous habits and large food intake, it primarily harms aquatic crops such as rice [9], which is the main alimentary crop in China. At the same time, the feeding of *P. canaliculata* is selective, endangering the species diversity of the aquatic plant community [10]. Its excretory–secretory products can pollute the water quality environment and contribute to water body eutrophication [11]. With characteristics such as strong adaptability, rapid reproduction, and also resistance to high temperature, hypoxia, cold, hunger, acid and alkali, water pollution, etc. [12,13,14,15], *P. canaliculata* more easily becomes the local dominant population, causing varying degrees of damage to the fish and shellfish resources in the water and endangering the local biodiversity. Meanwhile, as an intermediate host, it carries several main parasites that are harmful to human health and include *Echinostomarevolutum*, *Angiostrongylus cantonensis*, and *Gonathostomaspinigerum*, which cause a variety of serious diseases such as echinostomiasis, eosinophilic meningitis, gnathostomiasis, etc. [16,17]. Although *A. cantonensis* primarily uses *Achatina fulica* and *P. canaliculata* as intermediate hosts [18], eosinophilic meningitis caused by *P. canaliculata* is more common due to its large market sales volume, widespread distribution, and strong adaptability. In an outbreak of *A. cantonensis* in Beijing during 2006, as many as 160 patients became ill after eating undercooked *P. canaliculata* or related eatables, showing varying degrees of fever, headache, neck stiffness, and skin paresthesia [19].

Quantitative risk assessment of alien invasive species is the general trend of development. Currently, risk assessment is based on the analysis of the adaptability of alien invasive species in the target area. Species distribution modeling (SDM) has been widely used in assessing the risk of invasive alien species [20], as well as in simulating pest and disease spread [21]. It estimates its potential distribution in the target area, using species distribution data and environmental variables, and evaluates the importance of environmental variables using jackknife. Due to the accuracy of prediction, particularly in the case of few or incomplete distribution data, the maximum entropy (MaxEnt) model is one of the most commonly used models [22]. Furthermore, the kuenm package makes use of the flexibility of R and MaxEnt to allow for detailed model calibration and selection, final model development and evaluation, and extrapolation risk analysis [23]. The MaxEnt model optimized via kuenm could better predict the distribution of *P. canaliculata*, which will limit the future development of the species to some extent, with corresponding policies for prevention and control.

Since temperature is one of the most important factors that influence species diffusion and distribution, global warming threatens to accelerate the spread of invasive alien species [24,25,26]. In view of the restrictive effect of temperature on the expansion of *P. canaliculata,* we used environmental variables to estimate the suitable habitats for *P. canaliculata*, and the key environmental variables impacting the distribution were obtained. In this study, we generated a model of *P. canaliculata* distribution using MaxEnt optimized by kuenm, the global occurrence records of *P. canaliculata* and environmental variables were used to predict the potential distribution changes in the present and under four climate change scenarios in the future to aid in the prevention and control of its invasion and spread in China.

## 2. Materials and Methods

### 2.1. Environmental Variables

The environmental variables were downloaded from WorldClim version 2.1 (www.worldclim.org/data/worldclim21.html, released in January 2020) [27], with 2.5 minutes spatial resolutions, which include 19 bioclimatic variables, 36 climate variables (precipitation, minimum temperature, and maximum temperature in every month), and elevation. Under current and future conditions, the elevations were considered to be roughly the same. The current climate was represented by historical climate data from 1970 to 2000, and future environmental variables corresponding to the recent were divided into four periods with a 20–year interval from 2021 to 2100 to predict the future potential distribution of *P. canaliculata* under four shared socioeconomic pathways (SSPs): 126, 245, 370, and 585. Here, the Beijing Climate Center Climate System Model (BCC–CSM2–MR) was used for the global climate model (GCM) [28], which is considered to be appropriate for climate change in China [29,30].

ArcGIS version 10.7 (ESRI, Redlands, CA, USA, www.esri.com, accessed on 3 January 2021) was used to sample the data of distribution points in all environmental variable layers, where autocorrelation and multilinearity were inevitable and nonnegligible. To reduce the impact of collinearity, Pearson’s correlation analysis was performed on environmental variables. For model development, only variables with correlation coefficients less than 0.8 and eco-physiological significance were chosen. A jackknife test was used to determine variable importance, and variables with less than a 1% contribution were also eliminated. The results were analyzed using GraphPad Prism 8 software (San Diego, CA, USA, www.graphpad.com, accessed on 3 January 2021).

### 2.2. Occurrence and Analysis of Species

We obtained occurrence records of the *P. canaliculata* from the Global Biodiversity Information Facility (GBIF, www.gbif.org, accessed on 7 March 2021), and those occurrence records were deleted for which the values of the predictor variables were absent. To reduce spatial autocorrelation, the ENMTools package was used to delete duplicate occurrences in the same grid cell [31]. After filtering, this study compiled 405 occurrences of *P. canaliculata* (Appendix A).

Maximum training sensitivity plus specificity logistic threshold (MTSS) was used to convert the continuous MaxEnt predictions to presence/absence map, which was generally accepted as a promising method when only presence data were available [32,33]. The habitat of *P. canaliculata* was divided into four categories according to this value: Unsuitable habitat (0–MTSS), low suitable habitat (MTSS–0.4), moderately suitable habitat (0.4–0.6), and highly suitable habitat (0.6–1.0).

### 2.3. Change in Potential Distribution and Centroids

The tool “distribution changes between binary SDMs” of SDM toolbox version 2.4 [34] for ArcGIS was used to visualize the change of potential distribution and centroids in the present and under four climate change scenarios in the future. The binary map was carried out with MTSS as the boundary. For the distribution changes between binary SDMs, all binary maps under four climate change scenarios in 2021–2100 were compared with binary maps under the current climate, respectively. For the centroid changes, they were compared every 20 years in chronological order.

### 2.4. Optimization and Evaluation of Model

MaxEnt software (version 3.4.1) was used in this study to generate a model for the potential distribution of *P. canaliculata* since it has a better modeling effect even when there are fewer occurrence records [22]. We used a bootstrap with 10 repetitions to evaluate the predictive performance of annual models, with 75% data used as training and the remaining 25% used for testing.

To reduce the overfitting and complexity of the model, the kuenm R package was used to optimize feature combination (FC) and regularization multiplier (RM) [23], which are the most important MaxEnt settings that affect the model generation [35]. The model was tested with RM varying from 0.5 to 4 (0.5 increasing once), and all 31 possible combinations of five feature classes, including Linear (L), Quadratic (Q), Product (P), Threshold (T), Hinge (H). Model performance was assessed using statistical significance (partial ROC), omission rates (ORs), and the Akaike information criterion corrected (AICc) for small sample sizes. 

The model’s prediction effect was also assessed using the receiver operating characteristic (ROC) of the area under the curve (AUC), which would have a higher value when the species distribution deviates more from random distribution [36]. The evaluation criteria are as follows: AUC > 0.9 is considered excellent, 0.7 < AUC < 0.9 is considered good, 0.5 < AUC < 0.5 is considered acceptable, and AUC < 0.5 is considered invalid.

## 3. Results

### 3.1. Environmental Variables and Model Optimization

Among 248 candidates, only one statistically significant model met the omission rate and AICc criteria. In this candidate model (RM = 0.5 FC = LQP), the mean AUC ratio was 1.782, the partial ROC was 0, the omission rate was 0.05, and the AICc was 10991.12, which represents the lowest delta AICc after adjusting (Appendix A).

Pearson’s correlation coefficients among 11 main contribution variables are shown in Figure 1 and Appendix A. The relative contributions of the environmental variables to the MaxEnt model were estimated, and seven environmental variables, including Bio8, bio12, bio18, bio19, elev, prec3, and tmax11, were ultimately chosen for generating the model (Table 1).

### 3.2. Current Prediction of P. canaliculata

The potential distribution was predicted based on the above models and current environmental variables, which occurred in the south of the Yangtze River, as well as in the most southeastern part of China (Figure 2). The highly suitable, moderately suitable, and low-suitable habitats accounted for 5.62%, 8.486%, and 7.50%, respectively. The average AUC value of repeated operation was 0.962, and the standard deviation (SD) was 0.002, indicating that the model is highly reliable for the potential habitat of *P. canaliculata* and can effectively reflect its distribution in China (Figure 3).

According to the results of the jackknife test of variable importance (Figure 4), bio18 had the greatest influence on the distribution of *P. canaliculata*, followed by tamx11. The cumulative contribution of the two variables was more than 70%, which were major factors that contributed to the MaxEnt model.

The response curves showed how the predicted probability of presence changes as each environmental variable was varied (Figure 5). A probability value greater than MTSS indicated that the environment was suitable for the growth of *P. canaliculata*. As a result, the suitable range for the precipitation of the warmest quarter was 230.89~2044.43 mm, the suitable range for a maximum temperature of November was 7.86~33.17 °C, the suitable range for elevation was less than 607.44 m, the suitable range for a mean temperature of the wettest quarter was more than 13.96 °C, the suitable range for annual precipitation was 574.41~3803.38 mm, the suitable range for precipitation of March was 5.94~359.50 mm, and the suitable range for precipitation of the coldest quarter were less than 1076.88 mm.

### 3.3. Future Prediction of P. canaliculata

The change of *P. canaliculata* potential distribution from 2021 to 2100 in four SSPs of CMIP6 is shown in Figure 6 (Appendix A). This model predicted that global warming would promote the expansion of the potentially suitable habitats of *P. canaliculata*, with the total suitable habitats increasing (Appendix A and Figure 7). Furthermore, the centroid would move from south to north, particularly in SSP585 (Figure 8).

## 4. Discussion

In this study, the MaxEnt model was used to predict the potential distribution of *P. canaliculata* under current and future climatic conditions in China by using the worldwide distribution data. Despite the fact that *P. canaliculata* has invaded and colonized in China, the suitable habitat may be better predicted using global distribution data, especially given its strong invasiveness. The prediction was found to be reliable when compared with the current distribution of *P. canaliculata* in China [7] and the predicted AUC results (Figure 3). Our results showed that the suitable habitat range in China was wide, and it would continue to expand and move northward in the future.

*P. canaliculata* is one of numerous *Pomacea* spp. that are morphologically similar, and several of them have been brought to non-native regions and misidentified as *P. canaliculata* [37,38]. According to the results of sampling and sequencing, in China, these include primarily two species—*P. canaliculata* and *P. maculata* [7]. Indeed, we predicted the potential habitats of several other *Pomacea* spp. distributed in Asia such as *P. maculate*, *P. diffusa*, etc., none of which was as widely distributed in China as *P. canaliculata*. Therefore, it was considered that, while many species may be included, our result can still be illustrated.

When considering the transferability and prediction accuracy of the model, over–fitting and parameters selection were particularly important [39]. In this case, the kuenm R package was used to optimize the FC and RM in MaxEnt, reducing the model complexity while maintaining the model accuracy [23]. The background samples were generated by default, which may lead to inaccurate models for occurrence records clustering in better–surveyed areas [40,41]. Although we reduced the sampling bias using spatial filtering, it still should be mentioned and optimized in the next step of work. In addition to environmental variables, other characteristics such as the type and composition of water bodies, demographic data, and vegetation coverage played important roles in affecting the habitat suitability of species [42,43], which were insufficient in our research and require further investigation.

A previous study has found that low temperatures in winter play significant roles in limiting the growth of *P. canaliculata* [44]. While no eggs can overwinter, adults and juveniles may survive and contribute to the next reproductive season [45]. Meanwhile, *P. canaliculata* could boost the energy supply, tolerance ability, and supercooling to improve the cold resistance [46]. This could well explain why *P. canaliculata* has been found numerous times in Beijing, Shandong Province, and other northern Chinese cities in recent years. Therefore, it is critical to predict and prevent the northward movement of *P. canaliculata*. Consistent with the results of Byers et al. [47], the temperature in the coldest months and the amount of precipitation in the warmest months are the most important variables of all the environmental variables. However, because the research objects are different species of the same genus, and we used more environmental variables for screening, such as precipitation, minimum temperature, and maximum temperature for each month, the final variables differed. Although different SDM and GCM used in prediction, our results are basically consistent with Lv et al. [25] and Lei et al. [48], that is, *P. canaliculata* would continue to expand and move northward in the future, and temperature in the coldest months is critical climate variable. Given the high correlation among Tmax11, Tmin1, and Bio4, we chose Tmax11 for its high percent contribution and highest gain in the jackknife test. Tmax11 may also alter its overwintering ability due to the maximum temperature in the coldest months. Furthermore, we used the elevation variable in the model generation, and based on the jackknife test results, it decreased the gain the most when it was omitted, which therefore appeared to have the most information that is not present in the other variables. In addition, the results of the jackknife test showed that the environmental variable with highest gain when used in isolation is Bio18, which appeared to have the most useful information by itself. In general, our predicted results indicated that *P. canaliculata* grew in warm, humid and low–altitude environments, which was consistent with its growth habits.

Under the background of climate change, suitable habitats for *P. canaliculata* would shift to the high latitudes in the northwest, while suitable areas in the lower latitudes of the southeast would shrink, as high temperatures and global warming change the spread of this species [49]. The results of our response curve also revealed that an increase in mean temperature in the wettest areas will increase the risk of invasion and expansion, and with global temperature increase, some current humid areas could be transformed into arid areas [50], which may also be one of the reasons for its northward shift. Our potential distribution results in the future showed that under the condition of global warming, the low and moderately suitable habitats remained basically unchanged, and the highly suitable habitats noticeably increased, which was more obvious as the SSP grade increased. According to the results based on binary maps (Appendix A), the expansion area of suitable habitats increased with time in each scenario, and the contraction area increased at first and then decreased in SSP1 and SSP2 but kept raising in SSP3 and SSP5. The suitable distribution area differed under different SSPs, indicating that climate change has increased uncertainty about the suitable distribution of *P. canaliculata*. The difference among different SSPs was mainly due to the changes in greenhouse gas concentrations, especially the effect of CO_2_ concentration on temperature [51]. Taking a sustainable green road (SSP1) or middle of the road (SSP2) is the most effective way to slow down the invasion and expansion of *P. canaliculata.* On the other hand, the development of the use of high fossil fuel consumption will result in the obvious expansion and northward migration of suitable areas of *P. canaliculata*. Therefore, sustainable development will protect our environment, but it will also limit the colonization and spread of invasive organisms.

Physical, chemical, biological, and other methods are currently used to prevent and control invasive snails. Among them, chemical control is the most commonly used, and while side effects are obvious, it has the greatest impact on the original ecosystem, and there are issues such as drug resistance that limit its effectiveness [52]. Physical control necessitates a large amount of manpower, but the result is frequently insignificant, so it is frequently used as an auxiliary measure [53]. Biological control using natural enemies to reduce its species density is low cost and has strong sustainability, in addition to having little impact on the environment and species on the local level, which is the focus of research on how to prevent and control invasive snails [54]. Although there are still many problems to be solved in the research and application of biological control, it remains as one of the best long-term options for the prevention and control of invasive species in the long run, including research and application of natural enemies, parasitic natural enemies and microorganisms, plants, and local economic animals, and achieves better results through cooperative control by multiple means. Furthermore, many invasive snails are highly resistant to adverse conditions, such as pH and harsh environments [47], which may allow them to maintain a species advantage in the face of environmental deterioration, as well as having a greater impact on species diversity. Nevertheless, more comprehensive quarantine measures, as well as more effective biochemical control measures, should be implemented to prevent the further spread of *P. canaliculata* in China.

## 5. Conclusions

In this study, we used the optimized MaxEnt model to establish the current and future niche model of *P. canaliculata* in China. It was found that humidity in the warmest quarter and temperature in the coldest month play key roles in its growth, which was primarily related to its overwinter ability. With global warming, the invasive habitat of *P. canaliculata* could further expand and move northward in China. This study served as a resource for the management and prevention of *P. canaliculata* invasion. In addition, strict quarantine measures should be implemented in areas where *P. canaliculata* has not been reported according to the current research results, and appropriate biological, chemical and physical control measures should be combined to reduce the loss caused by *P. canaliculata* invasion.

## Figures and Tables

**Figure 1 biology-11-00110-f001:**
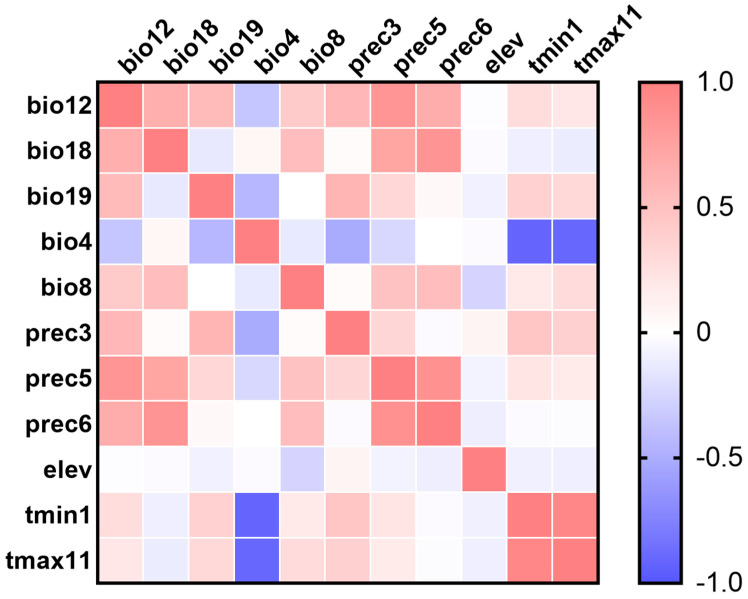
Pearson’s correlation matrix heatmap of environmental variables with contribution percentage greater than 1%. The variables include bio12 (annual precipitation), bio18 (precipitation of warmest quarter), bio19 (precipitation of coldest quarter), bio4 (temperature Seasonality (standard deviation × 100)), bio8 (mean temperature of wettest quarter), prec3 (precipitation of March), prec5 (precipitation of May), prec6 (precipitation of June), elev (elevation), tmin1 (minimum temperature of January), tmax11 (maximum temperature of November).

**Figure 2 biology-11-00110-f002:**
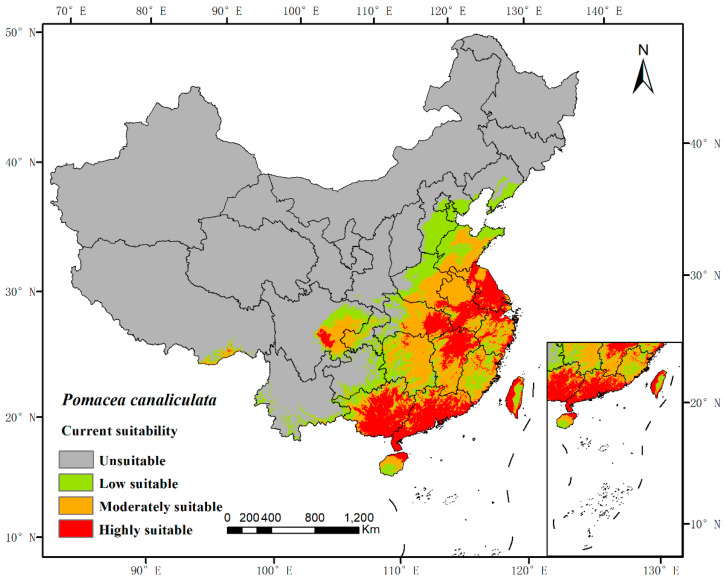
MaxEnt model predicted the current potential habitat suitability of *Pomacea canaliculata* in China. The base map was obtained from the Ministry of Natural Resources of China (http://bzdt.ch.mnr.gov.cn/index.html, accessed on 5 November 2021).

**Figure 3 biology-11-00110-f003:**
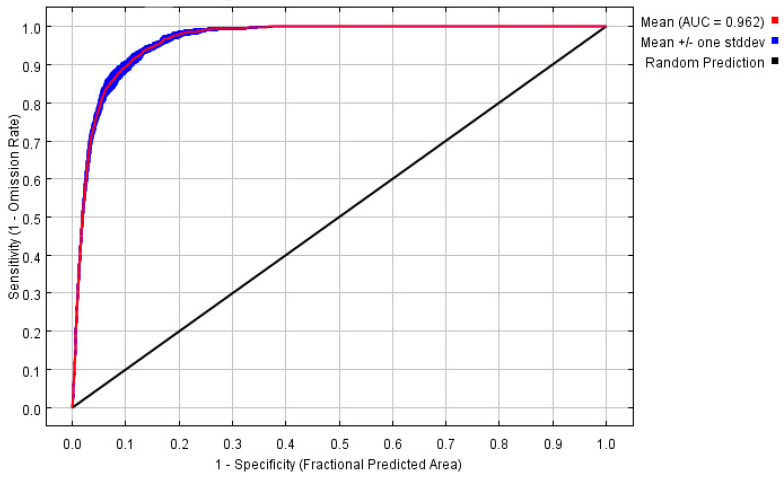
The receiver operating characteristic (ROC) curve and average area under curve (AUC) values for the optimized model over 10 replicate runs were shown in red, while blue margins show ± standard deviation (SD) calculated for 10 replicates.

**Figure 4 biology-11-00110-f004:**
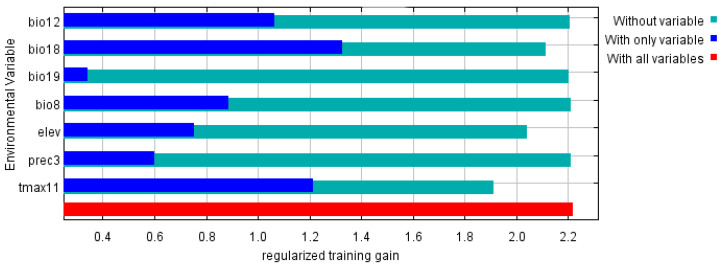
Jackknife test of variable importance in the *P. canaliculata* suitability distribution.

**Figure 5 biology-11-00110-f005:**
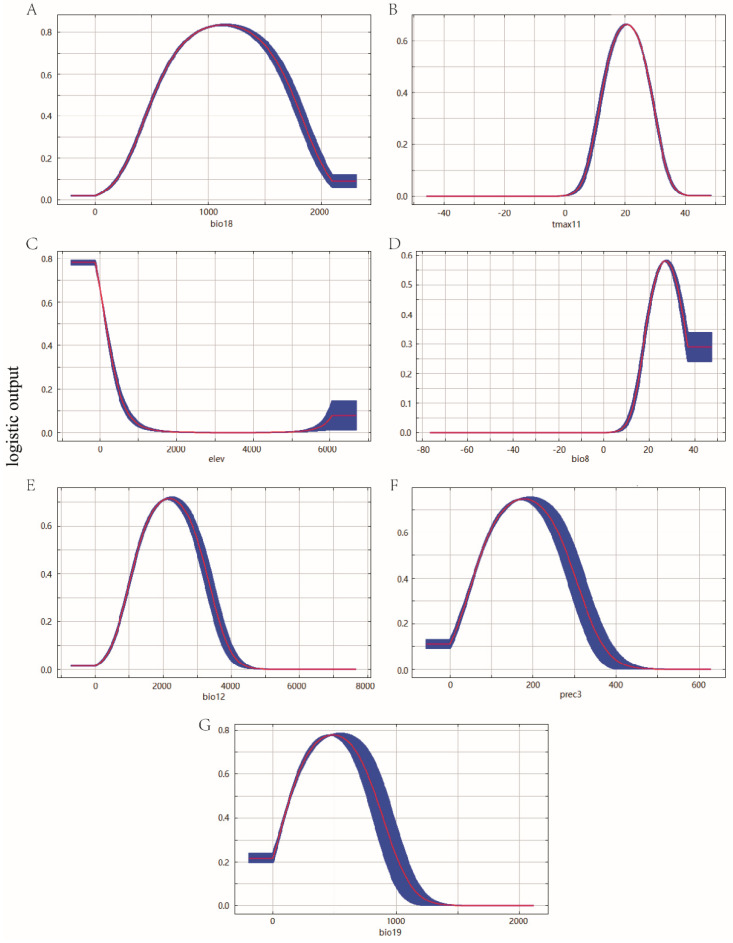
Response curves of environmental variables in the potential distribution model of *P. canaliculata*. The red curves represent average value over 10 replicate runs, while blue margins represented ± SD calculated for 10 replicates: (**A**) Bio18, (**B**) Tmax11, (**C**) Elev, (**D**) Bio8, (**E**) Bio12, (**F**) Prec3, and (**G**) Bio19.

**Figure 6 biology-11-00110-f006:**
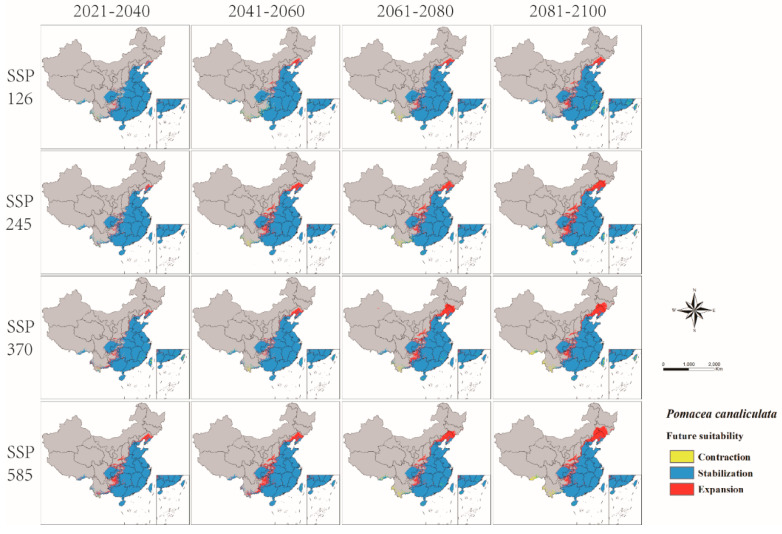
Changes in the potential distribution of *P. canaliculata* in China from 2021 to 2100 under four shared socioeconomic pathways (SSPs).

**Figure 7 biology-11-00110-f007:**
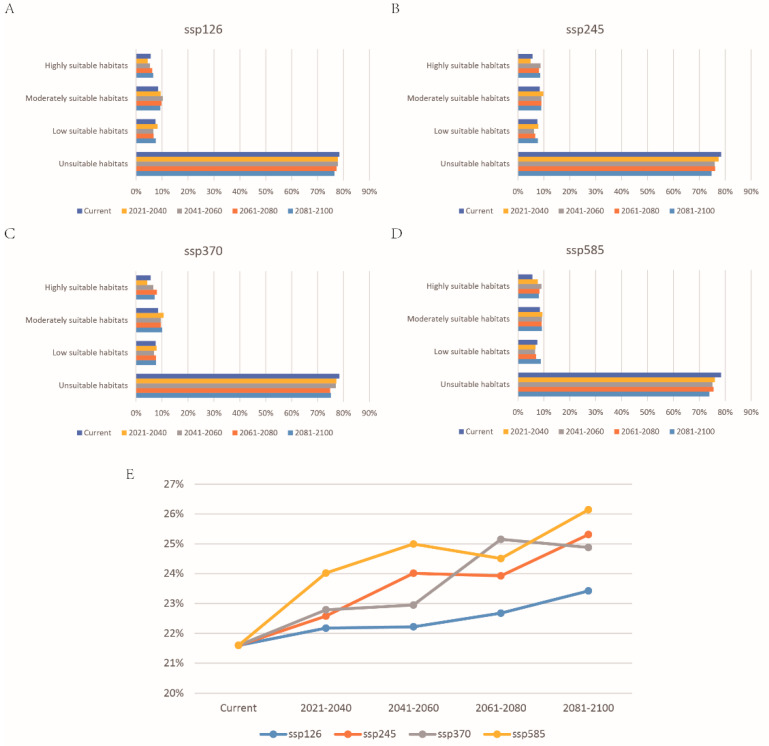
The proportion of suitable habitats for *P. canaliculata* under current and future climate change scenarios in China: (**A**) SSP126, (**B**) SSP245, (**C**) SSP370, and (**D**) SSP585; (**E**) change in the proportion of total suitable habitats (including low suitable, moderately suitable, and highly suitable habitats).

**Figure 8 biology-11-00110-f008:**
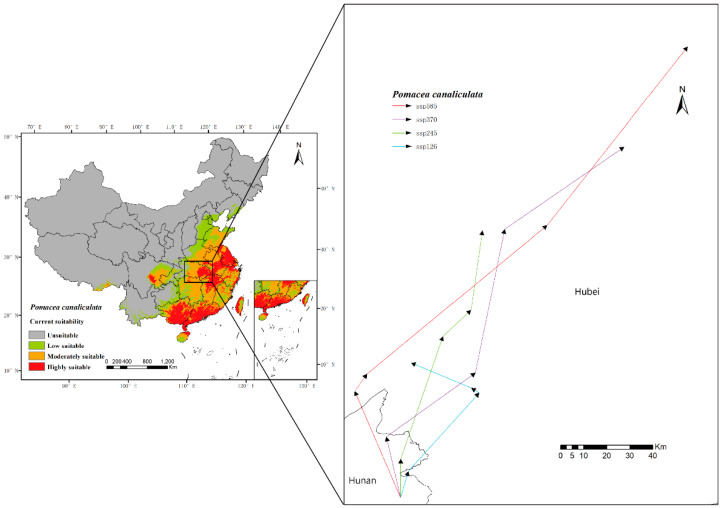
Change in distribution centroid under four SSPs in China from 2021 to 2100, with the distribution centroid shifting from the north of Hunan Province to Hubei Province.

**Table 1 biology-11-00110-t001:** Percent contribution and permutation importance of the environmental variables in the MaxEnt model.

Code	Environmental Variables	Percent Contribution	Permutation Importance
Bio18	Precipitation of warmest quarter	42.4	9.6
Tmax11	Maximum temperature of November	29.6	53.4
Elev	Elevation	17.6	19.2
Bio8	Mean temperature of wettest quarter	5.5	15.1
Bio12	Annual precipitation	1.9	0.5
Prec3	Precipitation of March	1.5	0.8
Bio19	Precipitation of coldest quarter	1.4	1.3

## Data Availability

Data are contained within the article or Appendix A.

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
