# Peer review of "Predicting Current Potential Distribution and the Range Dynamics of *Pomacea canaliculata* in China under Global Climate Change"

_biology, 2022, doi:10.3390/biology11010110_

Round 1
Reviewer 1 Report
Thank you for the opportunity to review your paper. I think you did a very good job of approaching selected techniques such as variable selection. However there are critical assumptions that were either not documented or not evaluated that would impact the model results and conclusions. I strongly suggest including more detail on how assumptions of spatial sorting bias, spatial stationarity, location quality, etc were all evaluated so the reader can determine the validity of the model results and conclusions.
I have included more detailed notes in the attached word document. In particular, spatial sorting bias and the assumption of spatial stationarity are often overlooked in published niche models. Again and again, scientists overlook spatial context when developing niche models (non-spatial regressions) to estimate spatial distribution: in particular, spatial autocorrelation and the fact that fitted trends can change across space. That error gets repeated over and over in the literature because it is so prevalent. I am hoping you will address these issues and set a better example in the published literature. You may also find more ecological insights to the model results as well.

Author Response
Dear reviewer:
Thank you for giving us valuable suggestions and comments about our manuscript, which have enabled us to further improve the quality and the conclusiveness of our manuscript Biology-1502694. Based on your comments, we have revised the manuscript. We hope that we have answered and clarified all questions and comments to your satisfaction and are looking forward to hearing from you soon.
Yours sincerely
Xuerong Li
Point-by-point response
Line 39. “Closer communication between countries is established as a result of globalization's facilitation.” While I think I understand what you are trying to say based on the rest of the paragraph, it might be best to rephrase so it has more impact as an opening statement. Consider the words “heightened connectivity” rather than “closer communication”. I think you could provide specific examples of human activity leading to introductions such as global trade, international travel/tourism, ballast water from ships, smuggling, etc. so the statement is less generalized.
Response: We have rephrased the paragraph 1 in introduction to emphasize the key points and provide some specific examples of human activity.
Lines 52-73. Excellent examples of impact statements which set the stage for why this invasive species has environmental impacts and is listed in the top 100 ICUN invasive species. Just before this paragraph, I think it would benefit the reader who isn’t familiar with this organism to describe it a bit more in terms of biology and native distribution. On line 46, it is cryptically stated that snails invaded China (harming its biodiversity) and I assumed you meant this organism. It might set the stage a bit better to give the common name of the snail after its scientific name, its native distribution, habitat requirements, etc. That way the audience can contrast what they know of its niche requirements with what you present in later analyses. Basically expand upon the first sentence on line 59.
Response: Following reviewer’s suggestion, we have added its common name, taxonomy and native distribution at the beginning of this paragraph in revised MS line 48, and the habitat requirements was supplemented in line 95.
Line 85. “will limit the specie’s85future development to some extent”. Do you mean that the model will constrain predictions to a more conservative estimate of future distribution?
Response: Our original intention is that if the distribution prediction is more accurate, the formulation of corresponding policies for prevention and control may better limit its future development. We have modified the sentence to express the meaning more clearly in line 94.
Line 88. This statement requires some citation or referencing my earlier suggestion of summarizing its habitat requirements.
Response: In revised MS, we have added its habitat to overwinter and more reference (doi:10.1111/j.1365-2427.2011.02579.x; doi:10.1541/ieejeiss.125.1552) to support this statement in line 95.
Line 119-123. Did you do any qualitative review of the data quality from GBIF? It is well known that GBIF data can report museum records as locations rather than field collected samples that represent true occurrence location. If not, you should report that uncertainty of the quality of the data locations may affect model outcomes.
Response: This is indeed a critical point. Our records are basis on Human observation and Material sample in GBIF, and we supplemented GBIF entries of our records in Supplementary Table S2.
Lines 124-128. This is an unusual method for determining the threshold. Usually, modelers will use criteria such as ROC metrics (sensitivity=specificity), minimum training presence, etc. Some papers to review:
https://doi.org/10.1016/j.ecolmodel.2008.05.015https://doi.org/10.1111/jbi.12058
Response: We agree with reviewer comment and reclassify the suitable habitat using maximum training sensitivity plus specificity logistic threshold (MTSS) as a threshold. All the results have been redone, and we hope it can further improve the quality of our MS.
Lines 130-147. There is no mention of how the background data or pseudo-absences were generated for the model. Please state how you generated the background samples (pseudo-absences). Did you let MaxEnt generate them by default? If so, this is extremely problematic as there is no correction for sample selection bias. Basically, the MaxEnt model (and every other niche model) assumes that samples are randomly selected and independently distributed throughout the study area. However, GBIF data records are a result of human activity that tends to cluster near populated areas, schools and museums, etc. So the presence data is spatially biased. If the pseudo-absence or background data is not filtered to mimic the spatial bias in the presence data, then the model will train on that spatial bias and produce results more aligned with human activity than environmental influence. This is a critical misstep in many published papers that use MaxEnt, when authors are not aware of the model assumptions.
https://onlinelibrary.wiley.com/doi/pdf/10.1111/ddi.12096https://www.jstor.org/stable/27645958https://www.stat.berkeley.edu/~wfithian/biasCorrection.pdf
Two strategies you can use. 1) use a kernel density estimator to estimate sampling effort and use that to stochastically allocate background samples. 2) Search GBIF for records of other species that were collected and use those locations as sampling effort where the observation is a zero for your target species (but is a one for different species). The last paper I cite above on target pooling describes this technique.
Response: We realized that background samples generated by default may lead to inaccurate models for occurrence records clustering in better surveyed areas. Since our limited research basis and time, we proposed it in our discussion in line 277. We will further study and apply these methods in our future work. Meanwhile, clustering was considered to not have a negative effect on Maxent performance (DOI:10.1139/Z08-078), and our results still have some scientific meaning to support the conclusion according to the sampling results recently.
Line 132. Cross-validation is a technique to assess whether the fitted model is robust to variation in the data (it is not overfitted). Another recommended technique is to assess whether your model is robust to spatial variation (i.e. does the model violate the assumption of spatial stationarity?). Consider running a spatial cross-validation where the samples are spatially subdivided across the study regions. Withhold a region for testing in the cross-validation to test for spatial sorting bias. Regular cross-validation has spatial autocorrelation between training and withheld samples, so the ROC comes back over-inflated (>0.90) You will find that a spatial cross-validation will be much less robust once spatial sorting has been addressed and may lead to insights about regionalizing your model over sucha large study area and whether some regions of China may respond differently to climate change.
https://doi.org/10.1111/1467-9884.00145 https://doi.org/10.1038/s41467-020-18321-y https://doi.org/10.1016/j.gecco.2019.e00894 https://doi.org/10.1111/j.1472-4642.2007.00344.x https://doi.org/10.1890/11-0826.1 https://cran.r-project.org/web/packages/blockCV/vignettes/BlockCV_for_SDM.html
Response: Thanks for your advice. Considered the spatial autocorrelation between training and withheld samples using cross-validation, we changed the replicated run type to Bootstrap, which have not intersection between the test set and the training set. We have redone the result and hope it can further improve the quality of our MS. We will further study the optimized cross-validation and apply these methods in our future work.
Results.
I cannot evaluate the validity of the results until the model assumptions have been addressed. The way in which variable importance is evaluated is fine. However, I think the authors need to discuss the results in coordination with model uncertainty (such as where the model extrapolates beyond the training data). I suggest using MESS maps (multivariate environmental similarity surfaces) to identify those regions where predictions are uncertain. You can see the extrapolation in the response curves in Figure 5. There are other ways to evaluate uncertainty too, which would help constrain conclusions on range shifts under climate change.
https://doi.org/10.1111/j.2041- 210X.2010.00036.x
https://rdrr.io/rforge/modEvA/man/MESS.html
https://www.ncbi.nlm.nih.gov/pmc/articles/PMC4278828/
Discussion/Conclusions.
It is likely that general trends will remain the same (moving northward) as that has already been documented for so many species in the literature. However, I strongly recommend that if this model is going to be used to understand vulnerable areas for invasion in China, to direct resources, etc please re-evaluate your model with regard to the assumptions I outlined and uncertainty. The areas being predicted are going to be different. If spatial cross-validation shows that some regions are statistically different in a global model, then regionalizing China into different models could provide some massive insights into the vulnerability of certain areas which may respond more quickly to climate change and have greater invasion potential.
Response: Thanks for your advices, we have redone all the results with the rule for replication and reclassify changing, and hope it can further improve the quality of our MS. We proposed some of your advice in our discussion and showed the deficiency. We still thought our results had some scientific meaning to support the conclusion according to the sampling results recently.
Reviewer 2 Report
Major comments:
- Pomacea canaliculata is one of many morphologically similar Pomacea species. Several of them have been introduced to regions outside their native range. In China, for instance, P. maculata and P. occulta can be misidentified as P. canaliculata. (see Hayes et al. 2012 doi: 10.1111/j.1096-3642.2012.00867.x, Yang et al. 2019 doi: 10.1002/ps.5241 and Yang & Yu 2019 doi: 10.6620/ZS.2019.58-13). The authors need to clarify how they validated the used GBIF records. If GBIF records were included without sufficiant validation, the modeling may not only be based on data from P. canaliculata but from multiple species with different ecological requirements.
- It remains unclear which records from GBIF have been included in the study. Respective GBIF entries need to be added to the records listed in Supplementary Table S2.
- The use of climate variables distinguished by month does not make sense given that the authors included P. canaliculata records from regions in both the southern and northern hemisphere, which experience opposite seasons. The maximum temperature of November, which was chosen for model generation, is not the same variable in different hemispheres.
- The rich body of literature on invasive Pomacea species has not been appropriately included in the introduction and discussion. Even studies that address very similar questions are ignored (e.g. Lv et al. 2011 doi: 10.1111/j.1365-2427.2011.02579.x, Lei et al. 2017 doi: 10.1007/s10661-017-6124-y).
- The manuscript is written in poor English and contains various flaws. Spelling, style, and grammar have to be checked and edited carefully.
Minor comments:
- Lines 46−49: Reference [3] is a paper on non-native species in marine and coastal habitats and not relevant regarding freshwater snails.
- Line 51: “Pomacea spp. is native to South America.” This is not correct, as some Pomacea species are native to North America (which includes Central America); see e.g. Cowie & Thiengo 2003 Malacologia 45: 41−100.
Lines 87−88: “Temperature is the main factor influencing species diffusion and distribution, and global warming could lead to its further spread and invasion to the north [24].” This statement is too general and not supported by the given reference.
- Line 108: The authors should include a reference for the global climate model used.
- Lines 174−175: “which may alter its overwintering ability due to the maximum temperature in the coldest months.” This interpretation does not belong in the results section.
- Line 178: “which was consistent with the growth habits of P. canaliculata” This is difficult to understand (and might rather belong in the discussion).
- Figure 6: The legends are too small. They should be replaced by a single larger legend.
Lines 199−202: “Because of the impact of global warming, humidity and temperature are more suitable for its growth, which is consistent with the growth characteristics of its preference for humid and warm environments, proving that the conclusion is credible.” This interpretation does not belong in the results section.
- Figure 8: The area on the map is difficult to locate. An overview map should be added.
- Line 239: “different genera of the same species” should be “different species of the same genus”
- Lines 275−278: “Biological control using natural enemies to reduce its species density is not only low cost and strong sustainability, but also has little impact on the environment and species in local, which is the focus of the research on how to prevent and control invasive snails[40].” Reference [40] is not about biological control but about chemical control using plant extracts.
- Line 285: “ocean acidification“ is irrelevant for a freshwater snail species
Author Response
Dear reviewer:
Thank you for giving us valuable suggestions and comments about our manuscript, which have enabled us to further improve the quality and the conclusiveness of our manuscript Biology-1502694. Based on your comments, we have revised the manuscript. We hope that we have answered and clarified all questions and comments to your satisfaction and are looking forward to hearing from you soon.
Yours sincerely
Xuerong Li
Point-by-point response
Major comments:
- Pomacea canaliculata is one of many morphologically similar Pomacea species. Several of them have been introduced to regions outside their native range. In China, for instance, P. maculata and P. occulta can be misidentified as P. canaliculata. (see Hayes et al. 2012 doi: 10.1111/j.1096-3642.2012.00867.x, Yang et al. 2019 doi: 10.1002/ps.5241 and Yang & Yu 2019 doi: 10.6620/ZS.2019.58-13). The authors need to clarify how they validated the used GBIF records. If GBIF records were included without sufficiant validation, the modeling may not only be based on data from P. canaliculata but from multiple species with different ecological requirements.
Response: Thanks for your advice. According to the results of sampling and sequencing in China, there are primarily two species, including P. canaliculata and P. maculate, while P. canaliculata distributed widely (doi: 10.1038/s41598-017-19000-7). Indeed, we have predicted the potential habitats of several other Pomacea spp. distributed in Asia such as P. maculate, P. diffusa and so on (as shown in the attachment which not redo in new methods), none of which was widely distributed in China like P. canaliculata, so it was considered that, while many species may be included, our result can still be illustrated. We also mentioned it in ms. discussion in line 266.
- It remains unclear which records from GBIF have been included in the study. Respective GBIF entries need to be added to the records listed in Supplementary Table S2.
Response: Following reviewer’s suggestion, we supplemented GBIF entries of our records in Supplementary Table S2.
- The use of climate variables distinguished by month does not make sense given that the authors included P. canaliculata records from regions in both the southern and northern hemisphere, which experience opposite seasons. The maximum temperature of November, which was chosen for model generation, is not the same variable in different hemispheres.
Response: On one hand, some variables of BIO1-19 have been shown to have spatial artefacts that could affect niche modeling, so we used more variables distinguished by month for prediction, on the other hand, our jackknife results show that Tmax11 had the most contribution among several high correlation variables, which appeared to have the most useful information by itself, so that we think it makes sense to use variables distinguished by month in our model to predict its distribution. Thanks again for your good advice.
- The rich body of literature on invasive Pomacea species has not been appropriately included in the introduction and discussion. Even studies that address very similar questions are ignored (e.g. Lv et al. 2011 doi: 10.1111/j.1365-2427.2011.02579.x, Lei et al. 2017 doi: 10.1007/s10661-017-6124-y).
Response: We have added it in our discussion in line 320. We have also supplemented some reference and details in our introduction to promote our MS.
- The manuscript is written in poor English and contains various flaws. Spelling, style, and grammar have to be checked and edited carefully.
Response: The revised manuscript has been checked and edited by a colleague who is a native English-speaker researcher. If necessary, it can be further retouched in the next revision.
Minor comments:
- Lines 46−49: Reference [3] is a paper on non-native species in marine and coastal habitats and not relevant regarding freshwater snails.
Response: In the revised MS, it has been replaced by an appropriate reference (doi:10.16250/j.32.1374.2018242.).
- Line 51: “Pomacea spp. is native to South America.” This is not correct, as some Pomacea species are native to North America (which includes Central America); see e.g. Cowie & Thiengo 2003 Malacologia 45: 41−100.
Response: We have changed it to “Pomacea canaliculata is native to North America” in line 48 and rephrased the sentence in line 57.
Lines 87−88: “Temperature is the main factor influencing species diffusion and distribution, and global warming could lead to its further spread and invasion to the north [24].” This statement is too general and not supported by the given reference.
Response: We agree with reviewer, and have added its habitat to overwinter and more reference (doi:10.1111/j.1365-2427.2011.02579.x; doi:10.1541/ieejeiss.125.1552) to support this statement in line 95.
- Line 108: The authors should include a reference for the global climate model used.
Response: We have added the reference in our revised MS (doi: 10.5194/gmd-12-1573-2019, 2019).
- Lines 174−175: “which may alter its overwintering ability due to the maximum temperature in the coldest months.” This interpretation does not belong in the results section.
Response: Following reviewer’s suggestion, we moved this part to discussion section in the revised MS.
- Line 178: “which was consistent with the growth habits of P. canaliculata” This is difficult to understand (and might rather belong in the discussion).
Response: Following reviewer’s suggestion, we moved this part to discussion section in the revised MS.
- Figure 6: The legends are too small. They should be replaced by a single larger legend.
Response: We have redone the picture and enlarged the legend.
Lines 199−202: “Because of the impact of global warming, humidity and temperature are more suitable for its growth, which is consistent with the growth characteristics of its preference for humid and warm environments, proving that the conclusion is credible.” This interpretation does not belong in the results section.
Response: Following reviewer’s suggestion, we have deleted it in our revised MS.
- Figure 8: The area on the map is difficult to locate. An overview map should be added.
Response: We have redone the picture and added an overview map to show the result clearly.
- Line 239: “different genera of the same species” should be “different species of the same genus”
Response: In the revised MS, it has been corrected.
- Lines 275−278: “Biological control using natural enemies to reduce its species density is not only low cost and strong sustainability, but also has little impact on the environment and species in local, which is the focus of the research on how to prevent and control invasive snails[40].” Reference [40] is not about biological control but about chemical control using plant extracts.
Response: In the revised MS, it has been replaced by an appropriate reference (doi:10.1002/ps.1424).
- Line 285: “ocean acidification” is irrelevant for a freshwater snail species
Response: We have deleted it in our revised MS.

Reviewer 3 Report
While various aspects of the world’s most dangerous invasive species (TOP 100) have been the target of multiple works, research on the range dynamics of the P. canaliculata in China is being studied for the first time. The authors use worldwide species occurrence records to predict the dynamics of species range dynamics under global climate change in time period 2021-2100. Although modern methods of species distribution modeling were used in the MS, nevertheless, improvement of the methodological part of the work is required. This is critical to support some of the statements in the final part of discussion. Therefore, I consider that the manuscript has scientific potential but requires substantial improvement and a second review to attain the standards of Biology.
Detailed comments - See attached file

Author Response
Dear reviewer:
Thank you for giving us valuable suggestions and comments about our manuscript, which have enabled us to further improve the quality and the conclusiveness of our manuscript Biology-1502694. Based on your comments, we have revised the manuscript. We hope that we have answered and clarified all questions and comments to your satisfaction and are looking forward to hearing from you soon.
Yours sincerely
Xuerong Li
Point-by-point response
Page 1, Lines 1-3
I think that from the title you need to remove “using optimized MaxEnt model” and “northward migration”.
For example, the following title is suggested – “Current potential distribution and predicting the range dynamics of Pomacea canaliculata in China under global climate change”
Response: We have changed the title to “Predicting current potential distribution and the range dynamics of Pomacea canaliculata in China under global climate change” in revised MS.
Simple Summary:
Page 1, Line 20
“Maximum entropy (MaxEnt) optimized by kuenm” is the methodical aspect. It should be replaced by “used species distribution modeling methods (SDM) ”
Response: Following reviewer’s suggestion, we have rephrased it in our revised MS.
Keywords
Page 1, Line 35
Only words that are not included in the MS title should be given.
Response: In our opinion, there could be an overlap between the keywords and the title, since they both highly summarize the content of the article.
Introduction
Page 2, Line 76
“Ecological niche modeling (ENM)” should be replaced by “Species distribution modeling (SDM)”, as this MS deals only with SDM. There is a difference between SDM and ENM (in general, SDM is not equal to ENM).
See:
Melo-Merino, S.M.; Reyes-Bonilla, H.; Lira-Noriega, A. Ecological niche models and species distribution models in marine environments: A literature review and spatial analysis of evidence. Ecol. Model. 2020, 415,
108837.
Petrosyan V, Osipov F, Bobrov V, Dergunova N, Omelchenko A, Varshavskiy A, Danielyan F, Arakelyan M. Species Distribution Models and Niche Partitioning among Unisexual Darevskia dahli and Its Parental Bisexual (D. portschinskii, D. mixta) Rock Lizards in the Caucasus.// Mathematics. -2020. -8 (8). https://doi.org/10.3390/math8081329
Response: We have replaced all the ENM with SDM in our revised MS. Thanks again for your good advices.
Materials and Methods
Page 3, line 99
Here is seems that such interpolation was performed by the authors, but this is already provided in WorlClim and acknowledged with a citation:
Fick, S.E. and R.J. Hijmans, 2017. WorldClim 2: new 1km spatial resolution climate surfaces for global land areas. International Journal of Climatology 37 (12): 4302-4315.
Response: Our statement may not be clear enough, so we rephrased this sentence and want to express that this part was downloaded directly from the website, where the URL is attached.
Page 3, line 108
Beijing Climate Center Climate 107 System Model (BCC-CSM2-MR) - here should be a citation:
Wu, T., Lu, Y., Fang, Y., Xin, X., Li, L., Li, W., Jie, W., Zhang, J., Liu, Y., Zhang, L., Zhang, F., Zhang, Y., Wu, F., Li, J., Chu, M., Wang, Z., Shi, X., Liu, X., Wei, M., Huang, A., Zhang, Y., and Liu, X.: The Beijing Climate Center Climate System Model (BCC-CSM): the main progress from CMIP5 to CMIP6 , Geosci. Model Dev., 12, 1573–1600, https://doi.org/10.5194/gmd-12-1573-2019, 2019.
In addition, an explanation must be given why this model was chosen from among the 40 CMIP 6 models, since for the BCC-CSM2-MR the ECS (long-term equilibrium climate sensitivity) indicator is approximately 3 ° C (see https://iowaclimate.org/2020 / 05/18 / 7c-global-warming-by-2100-cmip6-cranks-up-the-climate-sensitivity-estimate-for-cop26 /)
Response: Following reviewer’s suggestion, we have added this reference and given an explanation to choose this model (doi:10.1007/s13351-020-9204-9, 10.1007/s00382-016-3023-9) in line 119 in our revised MS.
Page 3, lines 109-115.
The order of data preparation for analysis is violated here. First you need to do spatial thinning of predictor variables using GraphPad Prism 8 software, and then determine the contribution of variables using MaxEnt. I recommend using only those variables that contribute more than 5%. It seems the text “11 main contribution variables above and results are shown in figure 1” should be presented in the results, and not in the section “Materials and Methods”.
Response: Following reviewer’s suggestion, we have rephrased this method, and the result remained the same. We still choose the variables with contribution more than 1% since we thought that Bio8 and Bio19 also play an important role in the model generation. Besides, we have moved figure1 to the result.
Page 4, lines 119-121.
“remove records that were outside the shapefile of the world map” - What does this mean -?
Response: Although some of the occurrence records have geographical coordinates, the data of each layer cannot be extracted from the ArcGIS, so they are removed when the distribution points are filtered, otherwise they will be deleted since they cannot be identified when the maxent is running. We rephrased this sentence in our revised MS.
Page 4, lines 121-123.
You have selected 405 occurrences records of P. canaliculata around the world, but have you checked the random distribution of these points? This is important because may have consequences on the results.
See:
Kaliontzopoulou, A.; Brito, J.; Carretero, M. A.; Larbes, S. & Harris, D. J. (2008): Modelling the partially unknown distribution of wall lizards Podarcis in North Africa: ecological affinities, potential areas of occurrence and methodological constraints. Canadian Journal of Zoology, 86: 992-1101.
Petrosyan V, Osipov F, Bobrov V, Dergunova N, Omelchenko A, Varshavskiy A, Danielyan F, Arakelyan M. Species Distribution Models and Niche Partitioning among Unisexual Darevskia dahli and Its Parental Bisexual (D. portschinskii, D. mixta) Rock Lizards in the Caucasus.// Mathematics. -2020. -8 (8). https://doi.org/10.3390/math8081329
Response: We reduced the spatial autocorrelation as much as possible using ENMTools. In the article you mentioned, it also said that “In contrast, clustering does not have a negative effect on Maxent performance, as sample size is more important”, so we mentioned this inaccurate possibility in our discussion line 276, and thought our results still have some scientific meaning to support the conclusion according to the sampling results recently.
Page 4, lines 124-128.
The use of a threshold value of 0.22 for identifying suitable habitats is unreasonable. A reliable method based on the continuous Boyes index should be used to determine the threshold.
See:
Hirzel AH, Lay GL, Helfer V, Randin C, Guisan A (2006) Evaluating the ability of habitat suitability models to predict species presences. Ecological Modelling, 199, 142–152. https://doi.org/10.1016/J.ECOLMODEL.2006.05.017
Response: We agree with reviewer comment and reclassify the suitable habitat using maximum training sensitivity plus specificity logistic threshold (MTSS) as a threshold in our revised MS, since it was generally accepted as a promising method when only presence data were available. All the results have been redone.
Page 4, lines 130-134.
It's good that you 75% data used as training and the remaining 25% used for testing. However, if a random distribution of all the species occurrence records points is not ensured, then this will affect the accuracy of the estimate of the suitability of the model.
Response: Thanks for your advice. As we mentioned before, we discuss this inaccuracy in our discussion in revised MS line 276.
Page 4, lines 143-147.
The value of AUC as model performance measure has been seriously questioned. One of the weak points is its dependence on the size, shape of the study area and etc. This analysis is flawed due to spatial autocorrelation. Records are not independent by spatially organized. There are now packages as Ecospat that allow estimating Boyce index. Boyce index lacks those drawbacks which has AUC index. It requires only data on species occurrence records and measures how much the predictive models differ from random distribution.
See:
Lobo, J. M., Jiménez-Valverde, A. and Real, R. 2007. AUC: a misleading measure of the performance of predictive distribution models. - Global Ecology & Biogeography 17: 145-151.
Broennimann, O., Fitzpatrick, M. C., Pearman, P. B., Petitpierre, B., Pellissier, L., Yoccoz, N. G., … Guisan, A. (2012). Measuring ecological niche overlap from occurrence and spatial environmental data. Global
Ecology and Biogeography, 21, 481–497.
Petitpierre, B., Kueffer, C., Broennimann, O., Randin, C., Daehler, C., & Guisan, A. (2012). Climatic niche shifts are rare among terrestrial plant invaders. Science, 335, 1344–1348.
Di Cola, V., Broennimann, O., Petitpierre, B., Breiner, F. T., D'Amen, M., Randin, C., Engler, R., Pottier, J. Pio, D., Dubuis, A., Pellissier, A, Mateo, R.G., Hordijk, W., Salamin, N., Guisan, A. (2017). Ecospat: An R package to support spatial analyses and modeling of species niches and distributions. Ecography, 40, 774–787.
Petrosyan V, Osipov F, Bobrov V, Dergunova N, Omelchenko A, Varshavskiy A, Danielyan F, Arakelyan M. Species Distribution Models and Niche Partitioning among Unisexual Darevskia dahli and Its Parental Bisexual (D. portschinskii, D. mixta) Rock Lizards in the Caucasus.// Mathematics. -2020. -8 (8). https://doi.org/10.3390/math8081329
Response: This is indeed a critical point. Although the value of AUC as model performance measure has been seriously questioned, we also evaluated the model using statistical significance (Partial ROC), omission rates (OR), and the Akaike information criterion corrected for small sample sizes (AICc) when we optimized this model. And ENMTools was used to reduce spatial autocorrelation, so we still think value of AUC have some scientific meaning to support this conclusion. We will further study this method you mentioned and apply these methods in our future work. Thanks again for your comments.
The manuscript does not provide methods for constructing SDM for various scenarios of climate change and period of modeling.
Response: In revised MS, we have added the methods for visualizing change in potential distribution and centroids in line 150.
Results
Page 7, line 198.
In the MS also does not describe which programs were used to determine the centroids of species range. It is recommended to present all the results after comparing the binary maps obtained under the conditions of the current climate and for various scenarios of climate change for the period 2021-2100 with a step of 20 years.
Response: We have supplemented the methods in line 150. At the beginning of our analysis, we drew the map with a step of 20 years, and found that some areas may frequently change between contraction and expansion in the comparison. Meanwhile, in view of the increasing inaccuracy of the comparison between the future and future, we still compare all the binary map with current, which is convenient for readers to see the changing trend of its distribution more clearly.
Discussion
In this section, it is recommended to discuss the results obtained using special indices (Gain, Lost, Change) based on binary maps, which allow to estimate the proportion and relative number of pixels (or habitat) lost, gained and stable for the time interval of modeling (2040, 2060, 2080, 2100).
Response: Following reviewer’s suggestion, we discussed the results based on binary maps in line 320 and supplemented the gain and lost areas in Supplementary Table S5. Thanks again for your advice.
Round 2
Reviewer 3 Report
In general, the new version of MS is much better than the previous one, however, there are minor comments. MS can be accepted after these comments have been corrected. These comments are presented below.
Page 3, line 104
Here it is important to add the exact URL and reference, i.e.
(www.worldclim.org/data/worldclim21.html)[Fick and Hijmans 2017].
Fick, S.E. and R.J. Hijmans, 2017. WorldClim 2: new 1km spatial resolution climate surfaces for global land areas. International Journal of Climatology 37 (12): 4302-4315.
This is important both for the journal Biology, as well as for the authors who created this very useful resource - BIOCLIM.
Page 3, line 115
It should indicate which version of ArcGis was used in the work, and a link to the ESRI developer
Page 3, line 126
This paragraph needs to be edited
…and ArcGIS was used to remove records whose data of each layer cannot be extracted from the ArcGIS, “that is, those occurrence records were deleted for which the values of the predictor variables were absent”
Supplementary Table S5, unfortunately this table is empty, i.e. in biology-1502694-supplementary.xlsx S5 tab contains no information
Author Response
Dear reviewer:
Thank you for giving us valuable suggestions and comments about our manuscript, which have enabled us to further improve the quality and the conclusiveness of our manuscript Biology-1502694. Based on your comments, we have revised the manuscript. We hope that we have answered and clarified all questions and comments to your satisfaction and are looking forward to hearing from you soon.
Page 3, line 104
Here it is important to add the exact URL and reference, i.e.
(www.worldclim.org/data/worldclim21.html)[Fick and Hijmans 2017].
Fick, S.E. and R.J. Hijmans, 2017. WorldClim 2: new 1km spatial resolution climate surfaces for global land areas. International Journal of Climatology 37 (12): 4302-4315.
This is important both for the journal Biology, as well as for the authors who created this very useful resource - BIOCLIM.
Response: We agree with reviewer comment and have added the exact URL and reference in our revised MS line 111.
Page 3, line 115
It should indicate which version of ArcGis was used in the work, and a link to the ESRI developer
Response: Following reviewer’s suggestion, we have added the version and link to the ESRI develporer in revised MS line 124.
Page 3, line 126
This paragraph needs to be edited
…and ArcGIS was used to remove records whose data of each layer cannot be extracted from the ArcGIS, “that is, those occurrence records were deleted for which the values of the predictor variables were absent”
Response: Thanks for your advice. We have rephrased this paragraph in line 138.
Supplementary Table S5, unfortunately this table is empty, i.e. in biology-1502694-supplementary.xlsx S5 tab contains no information
Response: The supplementary Table has been supplemented and re-uploaded.